# How Do We Select Right LLM for Each Query?

## Abstract

As Large Language Models (LLMs) continue to expand in both variety and cost, selecting the most appropriate model for each query is becoming increasingly crucial. Many existing works treat this as an offline problem, necessitating a data-gathering phase to compile a set of query-answer-reward triplets beforehand. They often struggle to determine the adequate number of triplets needed and are prone to overfitting if the data volume is insufficient. To address these limitations, we propose a new solution, the Multi-Armed Router (MAR), which applies multi-armed bandit theory—a perspective previously unexplored in this domain. Unlike previous works that base decision-making solely on regression techniques using static datasets (*i.e.*, constructed triplets), our method treats this as an online multi-LLM recommendation problem, which better mirrors real-world applications. Moreover, rather than the vanilla multi-armed bandit, our framework employs contextual bandit algorithms to navigate the trade-offs between exploring new models and exploiting proven models, while considering the dependency between the input query and the answer's reward. Due to the lack of an off-the-shelf dataset in this area, we construct WildArena, a dataset of 4,029 real-world user queries. For each query, there are seven open-ended responses derived from seven leading LLMs, respectively, with an evaluation score for each answer by using the LLM-as-a-Judge framework. We hope that the introduction of the new perspective and the dataset will facilitate the research in per-query LLM routing. Code and data will be made publicly available.

## 1   INTRODUCTION

The growing interest in Large Language Models (LLMs) has spawned a wide variety of models, each differing in performance and inference costs. These costs drive the need for frameworks that can assess both the performance quality and cost-effectiveness of these models effectively. The goal is to pinpoint the most suitable LLM for specific queries. Defined as the multi-LLM recommendation problem, this process involves selecting the most appropriate LLM for a query based on both performance quality and cost efficiency. This need stems from the considerable variability in LLM performance and cost for different tasks. For simple queries that require straightforward responses, less expensive, smaller LLMs are often sufficient, as the more complex outputs from larger LLMs might not offer extra value in such cases.

The complexity of inference can vary significantly between queries that represent much the same task. The query 'What is the asset value of Enron?' was simple to answer until August 2001, and the query 'Which country is Hawaii part of?' is simpler to answer than 'Which country is Antarctica part of?'. Static assignment of tasks to models, although simple, is unable to reflect the inevitable per-query variation in inference complexity.

The financial and environmental cost of inference on different LLMs varies by orders of magnitude. The difference is particularly stark when considering mobile platforms, where on-device models also have a significant latency advantage. As LLM inference inevitably becomes more common the associated costs will grow to the point where choosing the right model for a particular query will become essential.

One straightforward approach to this recommendation task involves collecting a large dataset of queries and their corresponding response ratings from various target LLMs. This dataset then serves to train a model, either through an end-to-end structure (Lu et al., 2023) or by using simpler architec-

tures such as a multi-layer perceptron (MLP) (Hu et al., 2024) to establish these associations. Once the training captures the essence of the routing function, the model is capable of making predictions for new incoming queries based on the trained model's insights. However, this process demands an exhaustive exploration of all potential queries and all LLM candidates, which is typically prohibitively costly. Moreover, due to the variability in real-world scenarios, the performance of these trained models on specific queries can often be erratic.

An alternative strategy is to improve response quality through augmentation. FrugalGPT (Chen et al., 2023), employs a cascade of LLMs ranging from smaller, less expensive models to larger, more costly ones, terminating the sequence when the response quality meets a satisfactory threshold. This ensures that simpler inquiries are answered using less costly models, thereby conserving resources. The method of (Li et al., 2024), in contrast, involves using a series of enhanced queries combined with the majority voting to derive a superior response. These methods can reduce the costs associated with inference and enhance response quality, but they often require multiple inference cycles, and they bear the associated costs.

A practical recommendation framework must be cost-efficient and capable of performing its task without access to the model's internal parameters. This enables the use of proprietary models and also requires the framework to dynamically adapt to a wide range of rapidly changing inputs. Traditional regression approaches often assume the presence of an ideal routing oracle and attempt to replicate it through regression on training datasets. The variable and unpredictable nature of real-world queries makes constructing a comprehensive static dataset difficult to achieve in practice.

Conventional static regression-based LLM recommenders struggle to identify the correct model in the face of the inevitable topic drift of real world queries. Dynamic augmentation approaches address this challenge but at the cost performing multiple inferences per query even when the same query is submitted repeatedly. Such approaches are often described in terms of the trade off between exploration and exploitation. The regression approach reflects a strategy of exhaustive static exploration, which is an extremum of the explore-exploit trade-off. The augmentation method does not use the results of previous queries to build a model to inform its decisions and thus represents an exploitation-focused strategy.

Our insight here is that the LLM recommendation problem mirrors the explore-exploit challenge (Thompson, 1933) that has been addressed repeatedly in developing other recommender systems. Exploration in this context corresponds to expending effort building a model that will predict the best LLM, potentially by gathering data through evaluating queries. Exploitation corresponds to performing inference on a recommended LLM to generate a response to a user query. The trade-off arises because building the perfect prediction model will waste effort, but a poor prediction model will recommend the wrong LLM. Multiple strategies exist that aim to perform only as much exploration as is required to support successful exploitation. By dynamically navigating between exploration and exploitation these methods avoid the pitfalls of both extremes(Schulz et al., 2018).

We propose the Multi-Armed Router (MAR), which employs a multi-armed bandit algorithm to balance exploration and exploitation in per-query LLM selection. One of the practical advantages of this approach is that interacts on a query level without needing to access any other aspects of the selectable LLMs. Our primary contribution consists of adopting this methodology and proving its advantages. These include a significant reduction in costs, enhanced flexibility, and increased usability compared to earlier methods. Future research could potentially delve into improved techniques for navigation between exploration and exploitation, along with the development of robust neural networks to boost recommendations.

We also examine the extensive diversity and substantial variability in subject matter that typify real-life settings. Previous investigations (Shnitzer et al., 2023; Hu et al., 2024) focused predominantly on evaluating specific datasets such as MMLU (Hendrycks et al., 2020), GSM8K (Cobbe et al., 2021), and MBPP (Austin et al., 2021), which encompass deliberately formulated questions aimed at particular domains, including undergraduate content, mathematics, and programming. Nevertheless, there is a discernible lack of datasets comprising queries with open-ended responses derived from real-world situations. To address this gap, we randomly selected 4029 queries from the chatbot-arena (Zheng et al., 2024) and WildChat (Zhao et al., 2024), which collect data through actual user interactions. We evaluated the responses using 7 LLMs from both open-source and proprietary sources of varying sizes and utilized LLM-as-a-judge to score the responses. This newly established dataset

is intended to contribute to the exploration of this field by providing an unprecedented collection of open-response data from multiple LLMs, complete with scoring.

Our contributions are twofold:

- We propose a Multi-LLM recommendation method that builds on the well developed multi-armed bandit approach to explore-exploit problems. This transition enhances our ability to balance exploration against exploitation while dynamically converging to the optimal routing strategy, significantly boosting overall performance and reducing costs.
- We introduce WildArena to address the shortage of datasets for developing Multi-LLM recommendation systems to tackle real-world open-ended queries. The dataset contains 4029 queries, alongside responses from both open-source and proprietary models, which are evaluated using the LLM-as-a-Judge approach.

## 2 RELATED WORK

### 2.1 CONTEXTUAL BANDITS

Multi-armed bandits (MAB) are often employed to address the explore-vs-exploit dilemma that commonly arises in various decision-making scenarios. Methods built on this framework vary from the classical $\epsilon$-greedy technique (Langford & Zhang, 2007) to advanced strategies such as UCB exploration (Kaufmann et al., 2012). The MAB model has been successfully applied in numerous areas, including recommendation systems and reinforcement learning. To overcome nonlinear problems and incorporate supervised elements, recent research (Collier & Llorens, 2018; Zhou et al., 2020; Zhang et al., 2020) has integrated neural networks within the MAB framework. This development allows for the implementation of more intricate and nonlinear functions for estimating rewards, a method that our study also adopts.

### 2.2 SUPERVISED LLM ROUTERS

In the field of LLM routing, researchers have explored different methods to improve efficiency and performance. ZOOTER (Lu et al., 2023), for instance, combines multiple LLMs, each specializing in a different area. The system uses a backpropagation of rewards to effectively train a routing function, which predicts the best LLM to use during inference, without needing to involve all LLMs. However, this method is limited to open-source models because it requires end-to-end training. Meanwhile, RouterBench (Hu et al., 2024) employs a multi-layer perceptron (MLP) to determine the indirect associations between queries and their respective response scores. This method is applicable to both open source and proprietary models, as it assesses the quality of responses using a structured scoring mechanism. Both ZOOTER and RouterBench require a thorough initial exploration of all possible LLMs resulting a heavy data collection phase. This not only leads to high costs but also restricts the adaptability of the systems, as they heavily depend on the initial exploration phase to adjust their exploitation strategies. This could result in unpredictable performance, especially when dealing with queries outside the initially trained domains. These challenges highlight the need for more dynamic and cost-effective solutions in the deployment of LLM routers.

### 2.3 RECOMMENDATION VIA QUERY AUGMENTATION

FrugalGPT (Chen et al., 2023) utilizes a cascaded model where responses are sequentially elicited from a hierarchy of language models, starting with smaller, less expensive ones and escalating to larger, more costly models. This progression is controlled by a trained scoring system that determines when the quality of the response is adequate to cease further inquiries. The success of this approach largely depends on the scoring mechanism's accuracy, and inevitably requires extra inference steps beyond the first appearance of the acceptable answer. As identified in LLM-as-a-Judge (Zheng et al., 2024), the evaluation capacity of these systems is often constrained by the performance limitations of the judging model, resulting in poor judgment capabilities for less sophisticated models.

(Li et al., 2024) introduces multiple prompts into a single query to generate diverse responses from an LLM, subsequently employing majority voting to identify the most satisfactory response. This method offers flexibility and does not require prior information. It is therefore potentially extensible

to multiple LLMs and real-time adjustments. It incurs a higher operational cost per query, however, and does not address the important case where the LLM is consistent in producing the wrong answer.

## 2.4 LLM Evaluation datasets

Various datasets exist for closed-ended evaluation, comprising queries with either a single or a collection of ground truth responses. Notable among these are MMLU (Hendrycks et al., 2020), GSM8K (Cobbe et al., 2021), and MBPP (Austin et al., 2021). The intrinsic requirement for defini­tive answers in these datasets typically restricts their topics to areas such as knowledge, facts, math­ematics, and programming. Such datasets fail to evaluate the capacities of different LLMs in per­forming tasks like composing email replies, commenting, or giving suggestions, which are critical to user interactions. In response, novel methodologies that utilize LLMs as evaluators have emerged (Zheng et al., 2024; Zhu et al., 2023b; Kim et al., 2023). These approaches validate the alignment of LLM judgments with those of human evaluators and facilitate extensive automatic scoring for response quality in open-ended real-world situations. This was a critical development in enabling more realistic open-ended evaluation, and part of the motivation for WildArena. This dataset em­bodies open-ended queries, tailored for a Multi-LLM recommendation system that aims to optimize the quality of responses to open-ended questions.

## 3 Method

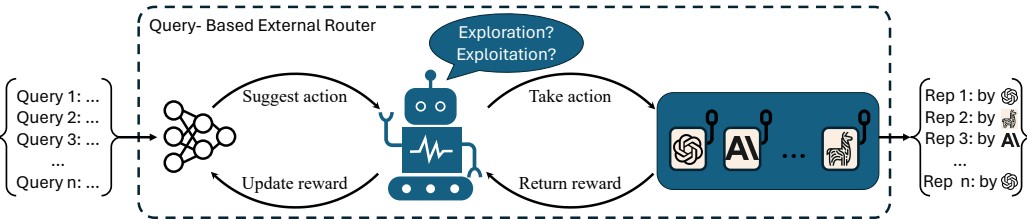

Figure 1: The Multi-Armed Router (MAR) paradigm that optimizes over the explore-exploit trade off in per-query LLM selection. When a query is input, a specialized neural network suggests a fitting LLM for processing. Integral to this strategy is a recommendation algorithm that acts as an internal controller, dealing with the balance between exploration and exploitation based on past decisions and outcomes, and implementing relevant actions. After choosing and using a specified LLM to handle the query, we analyze the response and receive a reward that assesses the effective­ness of the given response. This reward influences the subsequent update of the neural network. By consistently training the neural network to consult one LLM at a time, the system progressively evolves to guide each query to the optimal LLM.

### 3.1 Problem Setting

We begin by revisit the concept of the stochastic K-armed contextual bandit problem. Each inter­action in the decision sequence is denoted by $t \in T$, where the system is faced with $K$ distinctive feature vectors $\{\mathbf{x}_{t,k} \in \mathbb{R}^d | \ k \in K\}$. Upon this, the system selects an action $a_t$ and consequently receives a reward $r_{t,a_t}$. The overarching goal is to minimize the pseudo regret, defined as:

$$R_T = \mathbb{E}\left[\sum_{t=1}^{T}(r_{t,a_t^*} - \tilde{r}_{t,a_t})\right], \tag{1}$$

where $a_t^* = \arg\max_{a \in [K]} \mathbb{E}[r_{t,a_t}]$ represents the optimal action at round $t$ that maximizes the expected reward.

In the context of selecting among multiple LLMs for recommendations, each decision point $t$ cor­responds to choosing the most suitable LLM from a set of $K$ to address a given query $\mathbf{x}$. Herein, $a^*$ symbolizes the choice of LLM that offers an optimal compromise between performance metrics

such as quality and computational cost. Opting for any LLM other than the optimal induces an increase in $R_T$.

Following (Zhou et al., 2020; Zhang et al., 2020) we use neural networks as the basis of the representation to which the contextual multi-armed bandit algorithm is applied. Specifically, we employ a multi-layer perception (MLP)$f(\mathbf{x}; \boldsymbol{\theta})$ for each LLM or 'arm', defined as follows:

$$f(\mathbf{x}, \boldsymbol{\theta}_{a_i}) = f\Big(\mathbf{W}_n \cdot \sigma\big(... \cdot \sigma(\mathbf{W}_1 \cdot \mathbf{x} + \mathbf{b}_1)...\big) + \mathbf{b}_n\Big), \tag{2}$$

where $\sigma := max\{x, 0\}$ denotes the rectified linear unit (ReLU) activation function. The index $i \in K$ corresponds to the identified candidate LLMs for the recommendation. This learned MLP facilitates the approximation of the expected reward $\tilde{r}_{t,a_i}$. The actual reward received upon selecting an LLM $a_i$ is denoted as $r_{t,a_i}$, encompassing both performance and cost indicators:

$$r_{t,a_i} = g\big(s_{a_i}(\mathbf{x}_t), c_{a_i}(\mathbf{x}_t)\big), \tag{3}$$

In this relation, $s_{a_i}(\mathbf{x}_t)$ measures the quality of output produced by the LLM, while $c_{a_i}(\mathbf{x}_t)$ indicates the costs associated with using the selected LLM's API. The function $g(\mathbf{x})$ thus represents a balance between the quality of the output and the costs incurred, which we opt not to detail further. To mitigate the pseudo regret $R_T$, we have the equation:

$$Loss = MSE(\tilde{r}_{t,a_i}, r_{t,a_i}), \tag{4}$$

which uses an MSE loss to minimize the discrepancy between predicted and actual rewards following (Zhou et al., 2020).

### 3.2 THE MULTI-ARMED ROUTER

In Fig 1, we introduce the structural design of the proposed MAR. The primary objective is to direct each query to the LLM that will provide the best quality of response while minimising total inference costs. Naturally, MAR initially favours exploration so as to gather the information required to select the correct LLM. As more information is acquired MAR balances exploration and exploitation so as to minimize regret. The contextual multi-arm bandit is a well established solution to this challenge, and one that applies naturally to LLM recommendation. The resulting MAR algorithm is shown as Algorithm 1.

---

**Algorithm 1** The Multi-Armed Router

1: **Input:** A list of queries $q \in Q$
2: Initialize $\boldsymbol{\theta}_0$ as described (Zhang et al., 2020)
3: **for** $t = 1, ..., T$ **do**
4:     $x_t = Embedding(q_t)$
5:     **for** $i = 1, ..., K$ **do**
6:         Compute $\tilde{r}_{t,a_i} = f(\mathbf{x}_t; \boldsymbol{\theta}_{t-1,a_i}) + \Delta$
7:     **end for**
8:     Play $a_t$
9:     **if** $mod(t, batch) = 0$                                    ▷ Updates every batch step
10:         Receive reward $r_{t,a_t} = g\big(s_{a_t}(\mathbf{x}_t), c_{a_t}(\mathbf{x}_t)\big)$ where $a_t = \text{argmax}_a \tilde{r}_{t,a}$
11:         Let $\boldsymbol{\theta}_t = \text{TrainNN}\big(\{\mathbf{x}_i\}_{i=1}^t, \{r_{i,a_i}\}_{i=1}^t, \boldsymbol{\theta}_0\big)$
12:         Compute $\Delta$                    ▷ The calculation of $\Delta$ varies according to the chosen method.
13: **end for**

---

The Multi-Armed Router iteratively fine-tunes its routing strategy by leveraging feedback from observed rewards, striving to sustain the prescribed balance between exploration and exploitation. This balance is regulated by the $\Delta$ parameter, which can represent different settings like the exploration probability $\epsilon$, the upper confidence boundary in NeuralUCB (Zhou et al., 2020), or the $\sigma$ parameter in decision-making processes within Thompson sampling (Zhang et al., 2020). The parameter $\Delta$ thus

guides the system's behavior in selecting the appropriate degree of exploration required to inform an exploitation decision. By incrementally adjusting its routing decisions to reflect the unique features of each query and integrating accumulated historical data, the algorithm robustly improves its performance over time and decreases the proportion of explore events required to achieve accurate routing.

## 3.3 DATASET

A crucial component is the dataset employed to assess such an algorithm. An apt dataset for the Multi-LLM recommendation task encompasses a compilation of queries, various inferences from distinct LLMs addressing these queries, and the pertinent scores for each response. For evaluation purposes, continually invoking LLMs for inference and scoring is not feasible. Consequently, a pre-compiled dataset containing the inference results of LLMs is essential to simulate actual usage scenarios. The dataset that most closely meets these criteria at the moment is RouterBench (Hu et al., 2024), which aggregates queries from various established LLM benchmarks and conducts inference using 11 LLM models.

Nevertheless, the bulk of these inquiries predominantly revolves around scholarly knowledge, logical reasoning, mathematical challenges, and programming tasks. Such assignments usually come with precise, correct answers and are carefully chosen for their uniqueness and scarcity, which notably differs from the type and frequency of questions encountered in everyday user interactions. Conversely, solutions to questions from the real world often fall into a broad set, lacking a singular optimal answer. Consequently, we adopt a strategy proposed in (Zheng et al., 2024) where we utilize an LLM in a judging capacity, evaluating all responses collectively to establish a graded ranking of answers.

More specifically, we randomly selected $4,029$ queries from a pooled dataset combining chatbot-Anera (Zheng et al., 2024) and WildChat (Zhao et al., 2024), both of which are derived from real-world user interactions, and performed evaluations using seven LLMs recognized for leading performance in their respective scale among both open-source and proprietary variants.

The open-source models include: Qwen-32B (Bai et al., 2023) and Starling-7B (Zhu et al., 2023a).

The proprietary models comprise: GPT-3.5-0125, GPT-4-0125 (Achiam et al., 2023), Claude3-Opus, Claude3-Sonnet, and Claude3-Haiku (Anthropic, 2024).

As the judging model, we used Gemini 1.5 Pro (Reid et al., 2024) to eliminate any self-enhancing biases. We conducted five rounds of evaluation, averaging the scores to produce the final result. Further details on evaluating multiple queries concurrently are provided in the supplementary materials.

## 4 EXPERIMENTS

### 4.1 EXPERIMENT SETTINGS

Our approach primarily adheres to the steps detailed in Algorithm 1, utilizing NeuralTS (Zhang et al., 2020) as the contextual bandit framework to determine the $\Delta$ crucial for balancing exploration and exploitation. We adopt the hyper-parameters directly from NeuralTS.

We implemented our experiment using our WildArena dataset, the fundamental characteristics of which are illustrated in Table. 1. The Top 1 count shows the number of queries for which the given LLM achieves the highest score. For the sake of brevity, the reward vector $r_t$ has been converted into a one-hot encoding style. The Large Language Model (LLM) producing the highest reward is allotted a '1', whereas all other LLMs receive a '0'. The scores for GPT4 are adjusted by a $-0.06$ factor to prevent excessively strong LLM performance from leading to oversimplified solutions (such as routing all queries to GPT4).

Similarly, we've found that merely multiplying the cost by the performance in Equation 3 can become problematic. Even though performance scores range between $[0, 1]$, one LLM can be hundreds of times more costly than another, potentially leading to unhelpful solutions, even if they seem optimal in cost terms. While reward policies may differ across real-world scenarios, we opted not to investigate this specific factor in greater detail within the scope of our current research. Our

Table 1: Essential characteristics of the WildArena dataset are outlined. Notably, GPT4 exhibits superior performance relative to the alternatives. A detailed justification for this is presented in the appendix. For illustration purposes, we apply a $-0.06$ adjustment (noted with *) to the score of GPT4 to better align the scores. The unit of the price is US$ per million tokens input/output.

| | GPT3.5 | GPT4 | Claude3-Opus | Claude3-Sonnet | Claude3-Haiku | Qwen-32b | Starling-7b |
|---|---|---|---|---|---|---|---|
| Mean Score | 0.696 | 0.849 | 0.790 | 0.780 | 0.773 | 0.782 | 0.756 |
| Top1 Counts | 92 | 2022 | 486 | 456 | 296 | 365 | 312 |
| Top1 Counts* | 127 | 724 | 773 | 695 | 473 | 764 | 473 |
| Price | 0.5/1.5 | 10/30 | 15/75 | 3/15 | 0.25/1.25 | 0.8/0.8 | 0.1/0.1 |

core objective in this study, which focuses on multi-LLM recommendation systems, is to highlight the benefits of employing the recommendation approach specifically tailored for this particular task. Our method primarily draws comparisons with previous approaches that adopt an alternative methodology, offering insights into why our recommendation-based strategy presents a significant improvement.

## 4.2 ROUTING VS SINGLE MODEL STRATEGY

In Fig. 2, we offer a comparison of our proposed MAR with individual Language Learning Models (LLMs) that function without routing. Since the responses from the LLMs are the only element we need, this comparison is fair. In this analysis, we applied a 'pure-performance' cost policy, which means that only the performance of each LLM contributes to the reward function (represented as $c_{a_i}(\mathbf{x}_t) = 0$ in Equation 3). The results clearly show that, without any predetermined settings, MAR can develop a routing strategy that outperforms any standalone LLM, with lower regrets indicating better performance. MAR reaches optimal performance as early as step 2770, and the superlative performance continues to grow in the subsequent steps.

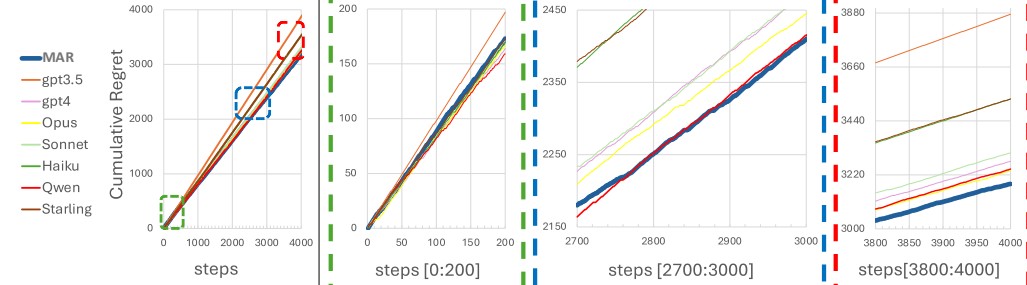

Figure 2: The graph depicts the collective regret over 4029 queries, comparing the MAR approach to a single LLM strategy. Certain sections of the graph are magnified for clearer illustration. Initially, in the first 200 steps, the MAR undergoes an exploration phase where its performance is comparatively lower. Between steps 2700 and 3000, MAR's regrets intersect and overtake those of the Qwen model. In the final 200 steps, MAR demonstrates marginally better performance than other strategies utilizing a solitary LLM.

The graph illustrating the MAR comparison with a single LLM, taking into account the cost implications, is depicted in Fig. 3. On the left side, the Starling-7b model is seen to predominantly lead the outcomes due to its price being 200 times lower, thus positioning it as the preferred choice. The MAR adjusts swiftly to this ideal scenario within 200 steps. The right side presents a more contextual real-world scenario where a new LLM is integrated into the MAR system periodically every 200 steps, following the sequence indicated on the plot labels. This setup tends to challenge traditional regression methods, yet it demonstrates MAR's ability to promptly fine-tune to an efficient routing option, irrespective of the LLM mix, without requiring prior information about any LLMs, thus offering considerable operational flexibility.

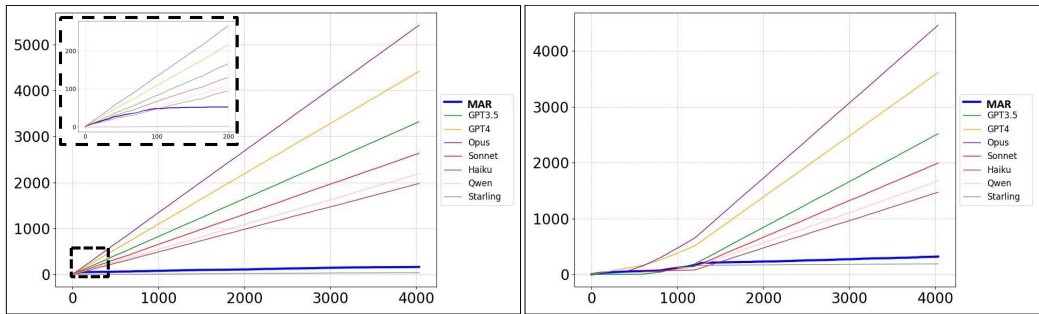

Figure 3: The left-side graph shows the cumulative regret plotted over steps using a policy that accounts for price. The MAR rapidly converges to the optimal strategy within 200 steps. On the right, the graph displays the LLMs that are introduced to the system sequentially, every 200 steps. It is evident that the MAR is able to quickly adjust to the incorporation of new LLMs while sustaining a minimal increase in cumulative regret.

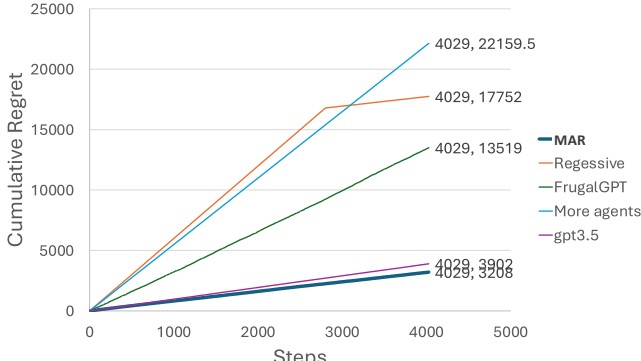

Figure 4: The comparison of the MAR approach versus traditional methods including the regressive techniques by Routerbench, cascade LLMs from FrugalGPT, and the query augmentation method by (Li et al., 2024)

### 4.3 COMPARISON WITH CONVENTIONAL METHODS

We also present conceptual comparisons of our MAR methodology against former techniques, as illustrated in Fig. 4. The depicted policy applies a 'pure-performance' cost metric. Moreover, the regressive approach incorporates a data-gathering phase where each received response must be evaluated using a scoring system to assess performance. Consequently, this method necessitates a comprehensive exploratory phase prior to any exploitation activities. The study titled 'More agents' (Li et al., 2024) employs an augmented query and voting mechanism. This method benefits from not requiring post-process scoring, which often depends on LLMs for evaluation, leading to a more gradual increase in cumulative regret compared to the regressive approach. Nonetheless, given that augmentation is required throughout the entire process, the cumulative regret eventually surpasses that of the regressive approach. In analyzing FrugalGPT under ideal conditions—where the scoring system flawlessly concludes exploration at the optimal LLM during sequential inference—it is observed that, despite favorable assumptions, the increase in cumulative regret remains significantly more rapid than that of our MAR method.

### 4.4 ABLATION STUDIES

We conduct ablation studies on MAR, using the final cumulative regrets as the performance metric, where lower scores indicate better results. The cost policy is set to 'pure performance'. We conducted experiments with various embedding models and different quantities of hidden channels in the MLP network shown in Table. 2. We evaluate the performance of two OpenAI embedding models: text-embedding-3-small (1536 dimensions) and text-embedding-3-large (3072 dimensions).

The results demonstrate that more powerful embeddings lead to superior MAR performance. Additionally, increasing the number of hidden channels in the MLP network from 100 to 1000 yields a modest improvement.

Table. 3 illustrates the impact of contextual multi-armed bandit methods on MAR performance. Given the highly non-linear nature of the context generated by the embedding model, neural network-based methods significantly outperform the linear method. Considering that the neural network employed in MAR is a simple 3-layer MLP, these experiments highlight MAR's potential, with ample room for further enhancements.

Table 2: Performance variations seen with different embedding models and neural network structures are measured by the final cumulative regrets after completing all the queries.

| Hidden channels | 100 | 1000 |
|---|---|---|
| text-embedding-3-small | 3208 | 3194 |
| text-embedding-3-large | 3154 | **3119** |

Table 3: The final cumulative regrets of various MAB methods were examined, with the neural network approach outperforming linear models.

| MAB methods | cumulative regrets |
|---|---|
| LinearUCB | 3425 |
| NeuralUCB | 3254 |
| NeuralTS | **3208** |

## 5 DISCUSSIONS

The MAR approach provides several benefits for Multi-LLM recommendation tasks, including cost-efficiency, quick convergence, adaptability, and practicality. Despite these advantages, the performance achieved by MAR is not as substantial as desired, falling short of the theoretical optimum, which would involve accurately routing all incoming queries.

We propose that this limited performance gain can be attributed to the inherent variability of the queries. In Fig. 5, we visualized the queries by sorting them based on the distance of their embeddings to their nearest neighbors. The red dots on the left represent the most closely clustered queries, while those on the right represent the most widely dispersed ones. Our analysis reveals that queries with tightly clustered embeddings exhibit significantly lower cumulative regret compared to those with more scattered embeddings. This observation implies that the available data may not provide adequate information for the router to make consistently accurate decisions. In such scenarios, additional data would be required for the system to operate effectively.

Given these limitations, MAR's cost-effective exploration-exploitation strategy offers a notable advantage over previous, more resource-intensive methods. This makes MAR a promising approach for Multi-LLM recommendation tasks, particularly in situations where data availability is limited or the cost of obtaining additional data is prohibitive.

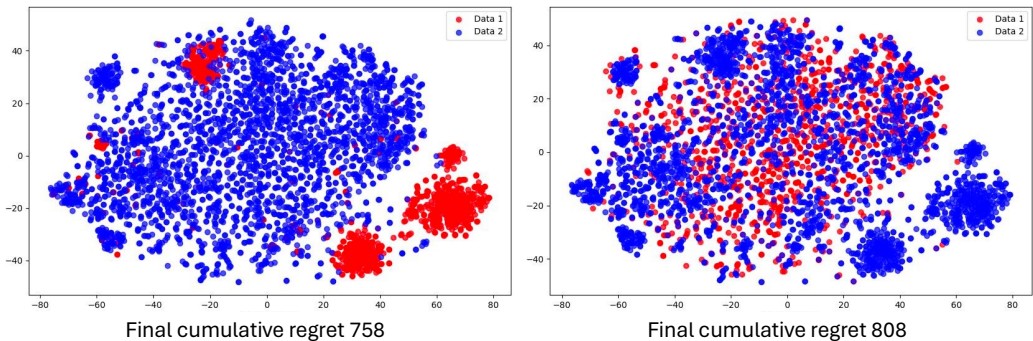

Final cumulative regret 758                    Final cumulative regret 808

Figure 5: The t-SNE visualization represents the embeddings, contrasting the initial 1000 queries (on the left), which exhibit the shortest distances to their closest neighbor, with the final 1000 queries (on the right) that display the greatest distances. The overall cumulative regret illustrates that MAR aligns more accurately with the left side than the right.

## 6   CONCLUSION

In summary, we introduce the Multi-Armed Router (MAR), a Multi-LLM recommendation method that utilizes the multi-armed bandit approach to optimize performance and minimize costs without accessing the internal elements of the LLMs. The multi-armed bandit approach is not only well understood theoretically, it allows explicit control over the explore-exploit trade off. Additionally, we present WildArena, a novel dataset comprising 4029 open-ended queries and their corresponding responses from multiple models, which aims to support the development and evaluation of Multi-LLM recommendation systems. Our study marks the first application of these techniques to the multi-LLM recommendation task, offering valuable insights into the advantages of this approach and paving the way for future research and advancements in this field.

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

# A APPENDIX

## A.1 THE WILDARENA DATASET CONSTRUCTION

### A.1.1 QUERIES COLLECTION

We gather queries from Chatbot-Arena (Zheng et al., 2024), which provides a dataset of roughly 30,000 queries, and from WildChat (Zhao et al., 2024), which includes 530,000 non-toxic conversations. To standardize the input data, we consider only the initial user interaction from each conversation as a valid query. As a result, the chosen queries do not compose coherent conversations. After filtering out duplicates and overly simplistic queries like "Hello" and "Hi," we create a representative sample by combining both datasets. With t-SNE, we visualize the query embeddings in Figure 6. The visualization shows that the Chatbot-Arena queries (red dots) are more dispersed and cover less area than the WildChat queries. This difference might be because the Chatbot-Arena queries were collected specifically for voting purposes, differing from real-world user scenarios.

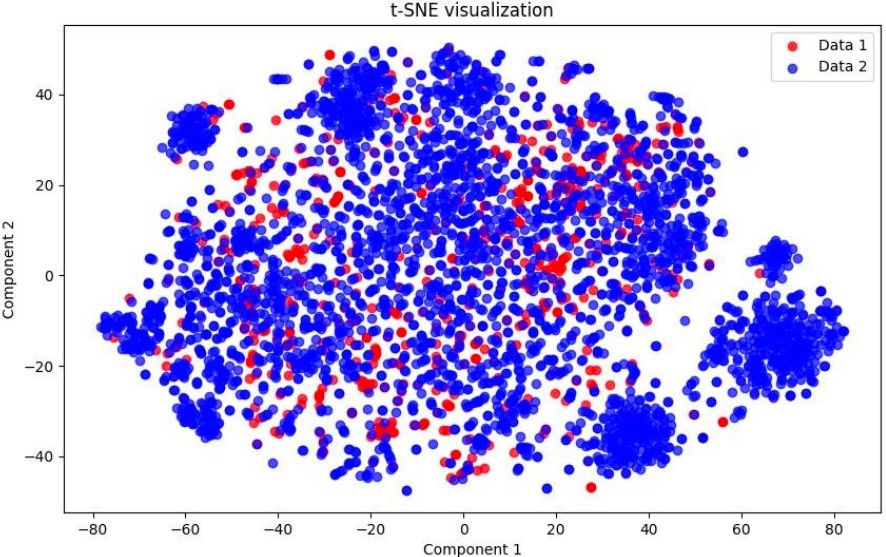

Figure 6: The t-SNE plot of the dataset. Red dots represent queries from Chatbot-Arena and blue dots represent queries from WilcChat.

After gathering the queries, we use seven LLMs to generate responses and record them. Following that, we employ the LLM-as-a-judge approach to assess these responses, which we will elaborate on in detail.

Given that there are seven responses to evaluate, scoring them individually would lead to significant variations. Therefore, we crafted a prompt that allows for scoring them simultaneously, as shown in Figure 7. This approach introduces three types of bias: positional bias, verbosity bias, and self-enhancement bias. To mitigate the positional bias, we randomize the order of the responses and repeat the evaluation five times, averaging the scores. Although verbosity bias generally favors longer responses, we found that the response lengths from the selected LLMs are relatively similar. As for self-enhancement bias, we use Gemini-1.5 pro, a Google model that directly competes with the other LLMs.

Initially, we collected 5,000 queries for our dataset. During the inference phase, however, we observed that many of these queries were flagged with "refuse to answer" responses. This was especially noticeable in the Claude3 models, which enforce strict policies that reject even mildly offensive or patent-related questions. Although we were able to filter out many problematic queries, some still slipped through. This partly explains why GPT-4 outperforms the Claude3 LLMs. As

```
prompt_start = (
    "As an impartial evaluator, your task is to evaluate the quality of answers from LLM in
response to user input, ensuring that your evaluation is unbiased and not influenced by the
sequence of responses or personal preferences, and output scores and very short reason in JSON
format. Each answer is to be scored out of a total of 1, with the consider of factors:\n\n"
    "- **Helpfulness**\n"
    "- **Relevance**\n"
    "- **Accuracy**\n"
    "- **Depth**\n"
    "- **Creativity**\n"
    "- **Level of Detail**\n\n"
)

prompt_question = "[Question]\n%question%\n\n"
prompt_answer = "[Answer]\n"
prompt_per_answer = "[Answer %rank%]\n%answer%\n\n"

prompt_end = (
    "[Output Format]\n[{\"score\": float, \"reasoning\": string}, {\"score\": float, \"reasoning\":
string}, ...]\n\n"
    "A-G Scores:"
)
```

```
# generating prompt
prompt = f"{prompt_start}{prompt_question.replace('%question%', query)}" + prompt_answer
    for index, (model_name, answer) in enumerate(answers, start=1):
        rank = chr(ord('A') + index - 1)
        prompt += prompt_per_answer.replace("%rank%", rank).replace("%answer%", answer)
    prompt += prompt_end
```

Figure 7: The default prompt for multiple entry comparison

previously mentioned, real-world usage patterns and frequencies can vary notably from the queries in Chatbot Arena's leaderboard, which are designed for voting purposes. This discrepancy helps clarify why our overall performance metrics differ from those on the leaderboard. Therefore, in our experiments, we adjusted down the GPT-4 by 0.06 to reflect a more realistic relative performance. Furthermore, as in our MAR, any customized cost-performance policy can be adapted to this structure. The reduced GPT-4 score can be seen as one such specific policy.

A.2 EXAMPLE OF A MULTI-RESPONSE SCORING

**Prompt:**

As an impartial evaluator, your task is to evaluate the quality of answers from LLM in response to user input, ensuring that your evaluation is unbiased and not influenced by the sequence of responses or personal preferences, and output scores and very short reason in JSON format. Each answer is to be scored out of a total of 1, with the consider of factors:

- **Helpfulness**

- **Relevance**

- **Accuracy**

- **Depth**

- **Creativity**

- **Level of Detail**

[Question] Was irrigation the primary or secondary to agriculture in the Persian Empire?

[Answer]

[Answer A]

Irrigation was a primary and essential component of agriculture in the Persian Empire.

Here are some key points about the role of irrigation in agriculture in the Persian Empire:

1. Arid climate: Much of the Persian Empire was located in an arid and semi-arid region, where rainfall was limited and unreliable. Irrigation was crucial to support agricultural production in these dry environments.

2. Extensive irrigation systems: The Persians developed and maintained highly sophisticated irrigation systems, including canals, dams, and qanats (underground aqueducts). These allowed them to tap into underground water resources and transport water over long distances to cultivate crops.

3. Centralized control and management: The Persian government played a key role in planning, constructing, and maintaining the irrigation infrastructure across the empire. This was necessary due to the scale and complexity of the systems.

4. Agricultural productivity: The reliable supply of water through irrigation enabled the Persians to cultivate a wide range of crops, including cereals, fruits, and vegetables, across diverse regions of the empire. This boosted agricultural productivity and supported the empire's economic strength.

So in summary, irrigation was absolutely essential and primary to the agricultural practices and productivity of the Persian Empire, rather than being a secondary or supplementary feature. The Persians' mastery of irrigation technology was a key driver of their agricultural success and the economic foundations of their vast empire.

[Answer B]

Irrigation was a crucial aspect of agriculture in the Persian Empire, playing a primary role in the development and success of its agricultural system. The empire, spanning a vast region with diverse climates, recognized the importance of water management for supporting agriculture, which in turn formed the foundation of its economy and society.

The Persians were known for their advanced irrigation techniques, including the use of qanats (underground water channels), which allowed them to tap into underground water sources and distribute it to farmland, especially in arid and semi-arid regions. This innovation enabled them to cultivate crops in areas where rainfall was insufficient, thus expanding agricultural productivity and supporting a growing population.

Agriculture, supported by irrigation, provided the empire with food, raw materials, and a basis for trade. It sustained the military, the administrative system, and the urban centers. The Persian kings, like Cyrus the Great and Darius I, actively promoted and invested in irrigation projects to enhance agricultural output and ensure food security.

In summary, irrigation was not secondary but rather a primary factor in the development and success of agriculture in the Persian Empire. It was an essential element that allowed the empire to thrive and maintain its vast territories by supporting a robust agricultural economy.

[Answer C]

Irrigation was a primary factor in the development and success of agriculture in the Persian Empire. The arid and semi-arid climate of the Iranian Plateau, where the Persian Empire was centered, necessitated the use of irrigation to support agriculture.

Key points:

1. The Persians developed sophisticated irrigation systems, including the qanat (underground aqueducts) and canals, to bring water from the mountains to the plains for agricultural use.

2. The qanat system, likely developed by the Persians, consisted of a series of vertical shafts connected by gently sloping tunnels that transported water from the water table at the foot of mountains to the lowlands.

3. These irrigation techniques allowed the Persians to cultivate crops in areas that would have otherwise been too dry, expanding the empire's agricultural potential and supporting its growing population.

4. The Persian Empire's agricultural success, supported by irrigation, was a key factor in its economic prosperity and political stability.

In summary, irrigation was not merely secondary to agriculture in the Persian Empire; it was a crucial primary factor that enabled the empire to develop a robust agricultural system in a largely arid region. The Persians' innovative irrigation techniques were essential to their agricultural success and, by extension, to the strength and longevity of their empire.

[Answer D]

Irrigation was primary to agriculture in the Persian Empire, as it played a crucial role in sustaining the empire's agricultural productivity and supporting its vast population.

The Persian Empire, which spanned from the 6th century BCE to the 7th century CE, covered a vast area that included modern-day Iran, Iraq, Egypt, and parts of Central Asia. This vast territory had diverse climates and geographies, ranging from arid deserts to fertile river valleys.

In areas with limited rainfall, such as Mesopotamia (modern-day Iraq) and Egypt, irrigation was essential for agriculture. The Tigris and Euphrates rivers in Mesopotamia and the Nile River in Egypt provided the necessary water for irrigation, allowing farmers to grow crops like wheat, barley, and dates. The Persian Empire developed advanced irrigation systems, including canals, ditches, and reservoirs, to distribute water efficiently across the agricultural lands.

Moreover, the Persian Empire's centralized administration and efficient tax collection system enabled the funding and maintenance of these irrigation systems. This, in turn, supported the empire's agricultural economy, which was the backbone of its wealth and power.

In summary, irrigation was primary to agriculture in the Persian Empire, as it enabled the cultivation of crops in regions with limited rainfall and contributed significantly to the empire's agricultural productivity and overall prosperity.

[Answer E]

The role of irrigation in agriculture during the Persian Empire is a topic that historians have studied extensively. Here's a brief overview of what we know:

Irrigation played a crucial role in agriculture in the Persian Empire, but it is difficult to say definitively whether it was primary or secondary. Both rainfed and irrigated agriculture coexisted in different regions of the vast empire.

In the arid and semi-arid regions, which covered a significant portion of the Persian Empire's territory, irrigation was essential for agricultural productivity. The Persians were highly skilled in developing sophisticated irrigation systems, including underground qanat systems, canals, and dams, to bring water from rivers, streams, and underground sources to the fields.

Regions like Mesopotamia, parts of modern-day Iran, and Central Asia relied heavily on irrigation for their agricultural activities, as the climate was too dry for successful rainfed farming. Crops such as wheat, barley, fruits, and vegetables were cultivated using irrigation in these areas.

However, in regions with more favorable rainfall patterns, such as parts of modern-day Turkey and the Caucasus, rainfed agriculture was more prevalent, and irrigation played a secondary or complementary role.

It's worth noting that the Persian Empire spanned a vast territory with diverse geographical and climatic conditions, so the importance of irrigation likely varied from region to region. Additionally, irrigation systems were often built upon or improved from earlier civilizations in the region, such as the Sumerians and Babylonians.

In summary, while irrigation was not uniformly primary or secondary across the entire Persian Empire, it was undoubtedly a critical component of agriculture, particularly in the drier regions, and a significant factor in the empire's agricultural productivity and ability to sustain its population.

[Answer F]

In the Persian Empire, irrigation was a critical secondary component to agriculture. Primary agriculture refers to the actual cultivation of crops and raising of livestock, while the term "secondary" in this context denotes the supporting or enabling activities that are fundamental to making primary agriculture effective.

The Persian Empire's territory included vast arid and semi-arid regions where rainfall was not sufficient to support agriculture. To overcome this limitation, the empire developed sophisticated systems of irrigation that included qanats (underground channels), canals, and dams. These technologies allowed water to be transported from mountainous regions to the plains for agricultural purposes, thereby enhancing crop yields and stabilizing food production.

Irrigation allowed the Persians to maximize the productivity of their lands, demonstrating their advanced understanding of hydraulic engineering. These irrigation systems were essential for the prosperity of the empire, enabling it to sustain a large population and contribute to the richness of Persian culture. Hence, while irrigation was not the primary activity of agriculture itself, it was of paramount importance to the success and expansion of agricultural practices within the Persian Empire.

[Answer G]

Irrigation was secondary to agriculture in the Persian Empire. Agriculture was the primary economic activity in the empire, and irrigation was a crucial method used to support and enhance agricultural productivity. Through the construction of extensive canal systems and dams, the Persians were able to effectively manage water resources and ensure a reliable water supply for their crops. However, while irrigation played a crucial role in supporting agriculture, it was not the primary economic activity in the Persian Empire.

[Output Format]

["score": float, "reasoning": string, "score": float, "reasoning": string, ...]

A-G Scores:

**Gemini-1.5 pro response:**

[

"score": 0.8, "reasoning": "Good explanation, but lacks depth and specifics about other agricultural advancements.",

"score": 0.85, "reasoning": "Very well written with examples and explanations, but lacks specifics and depth.",

"score": 0.9, "reasoning": "Provides detailed points on irrigation practices. Very thorough and accurate answer.",

"score": 0.75, "reasoning": "Lacks depth on irrigation practices specifically. More general Persian history.",

"score": 0.95, "reasoning": "Excellent nuanced answer. Covers multiple angles and perspectives on the topic.",

"score": 0.7, "reasoning": "Good summary, but lacks specific examples of irrigation techniques. Too brief.",

"score": 0.65, "reasoning": "Oversimplifies the complex relationship between irrigation and agriculture. Lacks depth."

]

The actual order of the above responses is : Haiku, Qwen, Opus, Starling, Sonnet, GPT4, GPT3.5.

