# OpenReview forum: "How Do We Select Right LLM for Each Query?"
_ICLR.cc/2025/Conference — ICLR 2025 Conference Withdrawn Submission_

### Official Review · Reviewer_xVxy · 2024-10-26

**Soundness:** 1
**Presentation:** 2
**Contribution:** 2
**Rating:** 3
**Confidence:** 4

**Summary:**

This paper proposes a Multi-Armed Router (MAR) based on contextual bandits in order to recommend the right LLM for a given query.
In order to test if their approach works, they compared the cumulative regret over a given number of queries and found that their router accumulates the least regret compared to other methods (e.g., just using a single model).

In conjunction to this work, the paper also introduced WildArena, a set of prompts with responses from different LLMs with their corresponding score using the LLM-as-a-judge framework. They used this as a test dataset for their work.

**Strengths:**

[S1] One of the main strengths of this paper is **formulating an online approach for query optimization (by routing to the right LLM at inference-time)**. I agree that building a router involves a lot of offline data collection and learning from past data, which can change over time—making the router outdated.
- [S1.1] I appreciate how this problem can also be seen in the lens of recommendation, which can open up more opportunities for further work.

**Weaknesses:**

Although the paper shows an interesting way to formulate the problem of choosing the right LLM for a given query, I have some concerns regarding the motivation, soundness, and presentation of the work:

[W1] **Clarity in the motivation.** It would benefit the paper to explicitly mention (in the introduction) the downstream use-case of their method in order to properly motivate the approach and the experiments. Is this LLM query optimization designed for (a) users querying an LLM on a daily basis? (b) For collecting data for instruction finetuning? (c) Or for collecting preference data (d) others?

- [W1.1] I also don’t entirely agree with the lack of data that necessitates creating WildArena. Again, this might be due to the lack of clarity in the motivation that’s why I’m still having trouble contextualizing the dataset. For example, [Ultrafeedback](https://huggingface.co/datasets/openbmb/UltraFeedback) has a similar set-up: different models with different ratings based on different aspects.

[W2] **Soundness of the experiments.** I have concerns on the robustness and reliability of the claims (soundness) based on the current experiments. For example:

- [W2.1] There are claims of cost-efficiency and practicality of the router, but I’m not entirely convinced because: (1) setting-up 7B and 32B models (as in their experiments) also incurs some running GPU costs and (2) recent improvements in API-based LLMs like OpenAI GPT-4’s Batch Inference has significantly reduced query costs. I suggest performing actual cost estimates (aside from cumulative regret) to strengthen this argument.
- [W2.2] Figure 4 shows that GPT-3.5 is almost on par with their approach in terms of performance. In practice, my opinion is that it's easier to set up an inference pipeline for a single model than having a committee of models just to get a 1% performance improvement (happy to be corrected). I suggest showing why it is still worth it to set-up a MAR framework and in what cases GPT-3.5 fails.
- [W2.3] The MAR framework uses the scores of an LLM-as-a-judge from WildArena as a target. However, experiments showing the validity of these scores are lacking. How do these correspond to human evaluation? What does it mean for a routing framework to perform well on WildArena?
- [W2.4] Section 4.2 Line 373 mentions that the MAR framework reaches the ideal scenario after 200 steps. How sensitive is this number on WildArena? I suggest strengthening this argument and show that this suboptimal regime is still small for different types of datasets.

[W3] **Nits: minor writing issues in the manuscript.** There are a few grammatical errors that need to be addressed in the manuscript. For example, the title might need to have an article “How do we select **the** right LLM for each query?” I’ve also noticed single-quotation marks incorrectly used when enclosing the text or usage of `\citep` instead of `\citet` or `\citealt` that breaks the flow of reading. These are minor nits and I encourage the authors to revisit the manuscript to update.

**Questions:**

I have integrated some questions in the Weaknesses section of this review. Below are some other questions that can help me understand the work further:

- [Q1] What types of examples were routed to “weaker” models and what types of examples were routed to stronger models?
- [Q2] Are there any measures of significance on Figure 2 and other cumulative regret computations?
- [Q3] Does this mean that in practice, the queries consumed during the exploration phase will receive suboptimal performance? Is it always in the first 200 steps?

---

### Official Review · Reviewer_H57F · 2024-11-02

**Soundness:** 3
**Presentation:** 2
**Contribution:** 2
**Rating:** 5
**Confidence:** 4

**Summary:**

The authors propose the MAR approach to dynamically select the best Large Language Model (LLM) per query by balancing exploration and exploitation. Additionally, it introduces the WildArena dataset, containing real-world user queries and responses from multiple LLMs, to support research on efficient multi-LLM routing.

**Strengths:**

The key advantages are its dynamic model selection approach, which optimizes the use of multiple LLMs based on each query, reducing costs while maintaining high response quality.

Besides, the introduction of the WildArena dataset addresses the gap in real-world, open-ended query data, which is beneficial to the community.

**Weaknesses:**

1. The model in the paper does not account for the evolving nature of LLM versions, where responses may change over time as models like "gpt4-turbo-latest" are updated to "gpt4-turbo." This limitation means that the model's recommendations may become outdated, as it does not adapt to these version changes, potentially affecting the quality and consistency of selected responses.
2. Even for the same model, responses to the same query can vary significantly (due to top-p sampling), leading to large fluctuations in reward scores. This raises a critical issue: how can the routing be reliably performed based solely on the query? Without accounting for the inherent variability in responses, the model’s routing decisions may lack consistency and reliability, potentially misguiding the selection of the most suitable LLM.
3.In the dataset construction process, the authors manually reduced GPT-4's scores by 0.06, which seems overly arbitrary and lacks sufficient justification. This manual adjustment introduces a bias into the evaluation process and may not accurately reflect the true performance differences between models, potentially skewing the results and limiting the generalizability of the findings.

**Questions:**

1. Have you considered any approach to address the issue of model version updates ("gpt4-turbo-latest" becoming "gpt4-turbo")? Given that model updates can lead to shifts in response behavior, are there strategies within the proposed framework to dynamically adapt to these changes over time, ensuring the routing decisions remain accurate and relevant despite evolving model versions?

2. In some scenario, the priority is not on cost-effectiveness but solely on selecting the best-performing LLM. Does your method account for this preference, and is it adaptable to situations where the primary objective is to maximize response quality？

---

### Official Review · Reviewer_62C7 · 2024-11-03

**Soundness:** 3
**Presentation:** 2
**Contribution:** 3
**Rating:** 5
**Confidence:** 3

**Summary:**

This paper proposes a method called Multi-armed router (MAR) to solve the problem of choosing a correct LLM for a specific query. The authors show that this method dynamically converges to the optimal routing strategy while saving cost at the same time.
Since during training responses from different LLMs for the same query are collected and graded using LLM-as-a-judge, they also provide the dataset consisting of around 4k queries and responses from 7 LLMs and their corresponding score.

**Strengths:**

- proposes a novel method that incorporates multi-armed bandit theory for training a router for different queries for LLM evaluation and shows that it is empirically effective

**Weaknesses:**

- In Figure 3, there is not axis label.
- see questions

**Questions:**

- some of the post-processing choices are not justified, e.g. why is the score of GPT4 adjusted by -0.06? If the criteria is made according to some target distribution of the scores, please explain.
- It is not clear to me why is updating the parameter \Delta related to exploration versus exploitation since in Algorithm 1 for each query responses from all K models are evaluated by the router.
- Since using LLM as a judge is not necessarily consistent depending on the task. Have you looked at the statistics of the ratings from the judge LLM and made sure they are normalized and not suffered from positional bias etc.
- The algorithm is presented poorly. In Line 10, if the output from an LLM is evaluated by another LLM, why is it that in the reward r= g(s,c), s is a function of the embedding of the query, but not the query itself? How is the embedding here uses?

---

### Official Review · Reviewer_DNMQ · 2024-11-04

**Soundness:** 3
**Presentation:** 2
**Contribution:** 3
**Rating:** 5
**Confidence:** 3

**Summary:**

The paper proposes a novel Multi-Armed Router (MAR) method that applies multi-armed bandit theory to address the problem of selecting the most appropriate Large Language Model (LLM) for each input query. Unlike previous regression-based approaches that rely on static datasets, MAR treats this as an online multi-LLM recommendation problem, which better mirrors real-world applications. The authors also introduce a new dataset called "WildArena" that contains 4,029 real-world user queries with responses from seven leading LLMs and scores evaluated using the LLM-as-a-Judge framework.

**Strengths:**

(1) The paper introduces a multi-armed bandit approach to the LLM recommendation problem, which allows the system to dynamically balance exploration and exploitation, leading to more efficient and adaptive model selection compared to static regression-based methods.
(2) The problem of selecting the right LLM for each query is an important and practical challenge as the number of available LLMs continues to grow. The proposed MAR method addresses this need in a cost-effective manner without requiring access to the internal parameters of the LLMs.
(3) The introduction of the WildArena dataset, which contains real-world open-ended queries and responses from multiple LLMs, is a valuable contribution that can facilitate further research in this area.

**Weaknesses:**

(1) The paper primarily focuses on the methodology and dataset introduction, but does not provide a comprehensive evaluation of the proposed MAR method compared to existing approaches. More detailed experiments, ablation studies, and comparisons with state-of-the-art methods would strengthen the paper's claims.
(2) The paper does not provide sufficient details on the implementation of the MAR algorithm, such as the specific contextual bandit algorithm used, the network architecture, and the hyperparameter settings. This may hinder the reproducibility of the study.
(3) The process of selecting queries and evaluating responses using LLM-as-a-Judge may introduce biases that should be acknowledged and discussed. The paper could benefit from a more detailed analysis of the dataset's characteristics and limitations.

**Questions:**

(1) Can you provide more details on the specific contextual bandit algorithm used within the MAR framework, such as the exploration-exploitation trade-off parameter settings and their impact on the performance?
(2) How did you ensure the diversity and representativeness of the queries in the WildArena dataset? What measures were taken to mitigate potential biases in the dataset?
(3) Can you compare the performance of the MAR method with other LLM recommendation approaches, such as ZOOTER and RouterBench, on the WildArena dataset? This would help readers better understand the strengths and limitations of your proposed approach.

---

### Note · Authors · 2025-01-15

I have read and agree with the venue's withdrawal policy on behalf of myself and my co-authors.